

# Software defined network intrusion system to detect malicious attacks in computer Internet of Things security using deep extractor supervised random forest technique

Muhammad Mujahid[1], Abeer Rashad Mirdad[1], Faten S. Alamri[2], Anees Ara[1] and Amjad Khan[1]

[1] Artificial Intelligence & Data Analytics Lab, CCIS, Prince Sultan University, Riyadh, Saudi Arabia
[2] Department of Mathematical Sciences, College of Science, Princess Nourah Bint Abdulrahman University, Riyadh, Saudi Arabia

Corresponding author
Faten S. Alamri,
fsalamripnu@gmail.com

## ABSTRACT

The architecture of software-defined networking (SDN) involves the separation of the network control plane from the routing plane. If this initiative turns out well, it has the potential to reduce operating expenses and the duration required to provide new services in comparison to traditional networks. However, this architecture has additional security concerns, including a single point of failure that could potentially provide any user with unrestricted access to the entire network. Nevertheless, it is essential to reduce the probability of security breaches. The development of immediate intrusion detection systems (IDSs) that can quickly spot and stop malicious activities like distributed denial of service (DDoS), DoS, web-attacks, and Bot-NET is an important part of SDN architecture. Several researchers are using cutting-edge methods, such as machine learning, to investigate and elucidate the causes behind the sudden rise in attacks and abnormal behavior, but the majority of these methods are deficient in terms of flexibility and accuracy. This study proposed a lightweight method for detecting different SDN attacks from intrusion-defined networks. The lightweight long short-term memory (LSTM) network has the capability to capture temporal patterns and sequential interactions in the SDN data. It also learned important context that is efficient for feature extraction and then developed supervised random forest (SRF) for the attack prediction. The dataset consists of 207,146 rows and 84 features that were preprocessed, including separate features and target attacks. The experiments show that the proposed method achieved 99.93% accuracy for attack detection and 0.0090 loss, confirming its efficacy. We also tested the proposed method on another SDN dataset and achieved 99.43% accuracy for multi-class attack detection. Furthermore, the use of supervised random forest reduces the model's complexity, resulting in increased overall efficiency.

## INTRODUCTION

Intrusion detection systems (IDS) show exceptional performance in two specific domains: detecting abnormal activities and monitoring network traffic. They are strategically placed on devices or across a network to analyze network traffic and detect signs of potential breaches. An IDS detects deviations from a preset usual behavior by searching for the signatures of known attack types (*Sowmya & Anita, 2023*). Afterwards, it notifies users about these irregularities and possibly harmful actions, allowing them to investigate at the software and protocol levels. The SDN architecture's flexibility, control, affordability, and dynamic nature make it a perfect fit for today's high-bandwidth applications. This architecture allows applications and network services to use the network without worrying about the infrastructure underlying it, keeping forwarding and network management duties separate (*Hurley, Perdomo & Perez-Pons, 2016*). Earlier versions of SDN highlighted the flexibility to build networks and the separation of control and data planes. Despite its numerous advantages, the solution to security issues may hinder the wider adoption of SDN. Typically, a security flaw in a network impacts only one node or segment. For instance, an SDN controller attack could compromise the entire network. Combining network resource management under one roof is one of SDN's many benefits. Implementing an SDN controller to manage network traffic and resources can enhance security (*Kreutz et al., 2014*).

SDN can now handle network traffic from Internet of Things (IoT) devices and out-of-date protocols with efficiency. SDN is susceptible to new kinds of attacks even if SDN provides advantages in IoT scenarios such as resource allocation, traffic management, and security intrusion detection and mitigation. The manufacturing process initially set up the Internet of Things devices, but the Mirai botnet malware later compromised their configuration (*Husnain et al., 2022*). These devices could remotely connect to a malicious server and issue orders through a botnet. Using these hacked bots, it is possible to flood the SDN controller with malicious network traffic. The goal of distributed denial-of-service (DDoS) attacks (*Alzahrani & Alenazi, 2023*) is to impede legitimate users from accessing certain websites by inundating them with a high volume of traffic originating from several computers with fraudulent IP addresses. Traditional IDS used the access frequency threshold to detect DDoS attacks. The existing flow table specifies that the OpenFlow SDN switch inside an SDN network will directly subject the intended recipients to harmful data. Because of this, the old IDS method is not very good at protecting OpenFlow OF SDN switches from DDoS attacks (*Xing et al., 2013*).

Traditional intrusion detection systems (IDS) often use deep packet inspection (DPI) to analyze packet headers and data payloads as a means to prevent virus assaults and denial of service attacks. SSH discussions frequently include data encryption *via* SSH. Advanced encryption techniques, such as dictionary attacks and brute force attacks, make it difficult for the DPI system to decrypt the packets and detect any malicious characteristics during the encryption of SSH packets. Therefore, using conventional DPI strategies will not effectively halt the attack (*Cepheli, Büyükçorak & Karabulut Kurt, 2016*). Software-defined networks are particularly vulnerable to specific repercussions caused by DDoS attacks

(*Manso, Moura & Serrão, 2019*). The SDN network management central controller may be more vulnerable to attack. SDN networks are susceptible to exploitation by attackers because of their high flexibility, which allows them to swiftly identify and exploit flaws. DDoS assaults against SDN networks may spread to other networks that use the same controller, leading to significant failures and disruptions (*Chen & Yu, 2016*). Intrusion detection systems (IDS) promptly halt any dubious or malevolent activity to safeguard networks and systems from compromise. They do this by consistently monitoring all activities, irrespective of their nature. IDS monitor system logs, network traffic, and other data sources to identify abnormal activity or signs of unauthorized access.

## Challenges

Many areas of research have concentrated on attack detection (*Zaman et al., 2025*), reliability, privacy and data security (*Wani, S & Khaliq, 2021*). Nevertheless, many classical models insufficiently represent critical sequential patterns, while many deep learning methods necessitate substantial network training, resulting in costly delays in attack detection and reduced affordability in real-time applications (*Adamou Djergou, Maleh & Mounir, 2021*). Consequently, it is important to employ advanced methodologies to investigate these issues. In a vast, dynamic environment, creating a security system that reliably discerns between normal and abnormal behavior is a challenging undertaking (*Deb & Roy, 2022*). The challenge in identifying features that may differentiate normal from DDoS attacks arises from the multitude of diverse datasets and different methods (*Khamaiseh et al., 2022*). The limited size of the training set results in a lack of low-frequency attacks (*Wang et al., 2024*). This limitation affects the evaluation of the proposed IDS using a real-world dataset that simulates DDoS, web attacks, and BotNet attacks. The high computational and resource requirements of many deep learning and conventional prediction models make network congestion in SDN settings more challenging. The performance of the real-time network may be degraded, particularly when there is heavy traffic.

## Problem statement and contributions

Traditional approaches in some cases unable to effectively detect temporal patterns in network traffic and experience significant overhead, particularly in dynamic settings such as SDN, leading to inadequate performance, especially against malicious or DoS attacks. Systems are unreliable and perform poorly due to the difficulty of tracking massive volumes of unclassified data. Despite its limitations, machine learning may be useful when integrated with NIDS (network intrusion detection system) to detect and classify threats. Our proposed model increases the accuracy of detection for different attack modes by using traffic trend analysis over time. It minimizes the number of false positives by using ongoing behavioral training to spot common and unusual patterns. Real-time attack identification is made easier by integrating the proposed model into SDN environment. Deep learning is a novel approach to estimating the likelihood of finding solutions to machine learning problems. The main contributions are as follows:

- The existing model has issues with complicated architecture, large batch sizes, high training processing, and slower SDN networks that lead to congestion. In contrast, our model is lightweight, uses fewer GPU resources, and provides a more accurate and adaptable detection method without causing excessive network congestion.

- The proposed methodology extracts temporal, long- and short-term flow behaviors with dynamic, complex network interactions, and the features are then fed into a supervised random forest model for detecting and mitigating DDoS attacks.

- To address a severe class imbalance across attack types, we employed Synthetic Minority Oversampling Technique (SMOTE), which significantly reduced misclassification errors and improved generalizations. Experimental results show that after using SMOTE, the classification errors went down from 5 to 0 for Brute Force Attack (BFA) attacks and from 6 to 0 for web attacks, proving that balancing the training data in SDN really works.

- We validate our proposed model by adding another intrusion dataset that has different types of attacks, showing that the model can work well in various intrusion situations. Furthermore, we employed an explainable AI technique to explain model decisions, improving reliability and transparency in security-critical environments.

- The proposed model enhances the detection rate of infrequent attacks and demonstrates superior performance compared to state-of-the-art models while minimizing false positive or negative rates. The proposed method for efficient SDN attack detection and classification is thoroughly discussed and evaluated.

## LITERATURE

The method involves placing an intrusion detection device node in each of the sub-networks created by SDN, which divides the network topology into numerous smaller networks. They enhanced a decision tree by using the black hole optimization approach to effectively identify intrusions in all sub-networks. This work proposes a deep learning-based approach to assault classification. After receiving the internal feature representations from the gated recurrent unit (GRU) deep learning layers, the model used principal component analysis (PCA) to identify the most suitable features. After feature fusion, the fully connected network is prepared to detect and classify threats. The proposed feature-fused GRU network outperformed both the GRU model and other popular, industry-standard models that rely on machine learning (*Ravi, Chaganti & Alazab, 2022*).

The authors created a network that used BiLSTM and CNNs to improve network intrusion detection through multiclass and binary classification. They assessed the performance of the proposed model using the two most widely used datasets, UNSW-NB15 and NSL-KDD. In addition, they extensively used the InSDN collection, which is specifically available for SDN. Here, it is evident that the suggested model was successful; they achieved outstanding results with much reduced training time (*Said, Sabir & Askerzade, 2023*).

The authors showcased a completely innovative and reliable technique for transmitting data that is also capable of detecting attacks on IoT devices. They employ a clustering method that uses active nodes' motion patterns to partition the network into many

regions. This system equips each component with a controller capable of exchanging security rules with other sections. The controller node, which serves as the central point of contact for all traffic exchanged between nodes within the subnet. Consequently, each management node consistently utilizes an EL model to analyze data on network activity and detect hazards (*Rui, Pan & Shu, 2023*).

Another study's goal was to provide a secure base for the IoT healthcare ecosystem using SDNs. The authors proposed a hybrid approach that combines deep learning and machine learning techniques, specifically employing DNN in conjunction with SVM, to identify instances of network breaches in healthcare data collected from sensors. Moreover, this system has the ability to efficiently monitor connected devices and suspicious activity. Finally, used a range of performance metrics to evaluate the efficacy of the proposed architecture in relation to healthcare application scenarios. The test results make it clear that the suggested strategy is better than the most cutting-edge methods used today to find and stop attacks in SDN-enabled IoT networks (*Arthi, Krishnaveni & Zeadally, 2024*).

The research conducted a comprehensive examination of machine learning techniques for SDN intrusion detection. The presentation discussed several machine learning techniques and approaches for detecting intrusions, and then proceeded to explain the architecture of SDN. They discussed the benefits of using SDN. They showcased state-of-the-art research on machine learning methods for detecting intrusions in software-defined networking. They provide a comprehensive overview of several studies, including a detailed examination of the advantages and disadvantages of each. Finally, they examined and emphasized significant research inquiries and probable future directions for using machine learning to detect software-defined networking intrusions (*Kumar & Alqahtani, 2023*).

The proposal recommends the implementation of a specialized HFS-LGBM-IDS for SDN. They used two-phase hybrid feature selection technique to get the optimal feature group and minimize the data size. The studies conducted on the NSL-KDD dataset demonstrate that the proposed system outperforms existing approaches (*Logeswari, Bose & Anitha, 2023*).

In the framework of SDN, the authors propose a method known as NIDS-DL. The suggested strategy combines a number of deep learning techniques with NIDS. The approach uses a feature selection technique, choosing 12 features from the total of 41 features in the Network Security Laboratory Knowledge Discovery Databases (NSL-KDD) dataset. Among the various classifiers we used were convolutional neural network (CNN), deep neural network (DNN), recurrent neural network (RNN), long short-term memory (LSTM), and gated recurrent unit (GRU). The method produced accuracy ratings of 98.63%, 98.53%, 98.13%, 98.04%, and 97.78%, respectively, when comparing the classifiers' scores. To get the best results, the Network Intrusion Detection System Deep Learning (NIDS-DL) method preprocesses the dataset and applies five deep learning classifiers. The recommended method was successful in binary classification and attack detection, indicating that NIDS-DL has a tremendous deal of promise for future applications (*Hadi & Mohammed, 2022*).

The purpose of this research was to provide Secured automatic two level Intrusion Detection System (SATIDS), which was an updated LSTM network, in order to differentiate between malicious and non-malicious traffic. It also specifies the type of high-performance subattack and the category of attacks in question. To train and evaluate the proposed method, make use of two of the most current actual Internet of Things datasets, namely Telecommunications and Networking Internet of Things (ToN-IoT) and Intrusion dataset for Software Defined Networks (InSDN). They analyzed and compared the proposed system's performance to other IDS to demonstrate its effectiveness (*Elsayed et al., 2023*).

Employed a deep learning algorithm to evaluate the efficacy of our DeepIDS in detecting network intrusions. The results suggest that the approach showed potential for future progress. Validated the potential of deep learning in the context of flow-based anomaly detection system by conducting a comparative analysis with other classifiers. The deep learning-based IDS technique shows potential in a SDN environment (*Tang et al., 2020*).

To train deep learning models to identify potentially malicious or suspicious packets, the SDN switch has sequentially logged the packet lengths. The proposed model uses and contrasts four distinct deep learning models: CNN, LSTM, multilayer perceptron (MLP), and a stacked auto-encoder. The experimental results indicate that the recommended MLP-based DL-IDPS obtains the highest accuracy (*Lee, Chang & Syu, 2020*).

The authors of this article developed a deep learning-based method for predicting network assaults in SDN-IoT networks. Using a four-hidden-layer model based on LSTM, identify attacks in the SDN-IoT network dataset and categorize them into several types. After comparing the DL model's performance to that of the ML classifier SVM and the other two DL models—DNN and CNN—we found that it outperformed all three in identifying the various network hazards present in the dataset (*Chaganti et al., 2023*). To identify assaults, researchers put forward and evaluated a DCNN model in an environment of software-defined networks (SDNs). They make use of the InSDN dataset (*Hnamte & Hussain, 2023*).

*Said Elsayed et al. (2020)* presents a novel approach to detecting anomaly-based attacks in an imbalanced dataset using LSTM autoencoders and SVM. The training data only contains samples that are typical of the classes. During our research, we developed a model that accurately determines and extracts the most relevant qualities from the provided parameters, as referenced in. Afterwards, the model utilizes deep learning techniques to classify invasions. Particularly important is the fact that it is not accurate to think of the underlying data points as samples from a single distribution. They have their roots in two distinct distributions, one of which is specific to a particular domain and the other of which is applicable to all network invasions. Both of these distributions contributed to their development (*Elsayed et al., 2021*).

*Singh et al. (2018)* proposed an approach to protect SDNs against distributed denial of service attacks is machine learning. There are three parts to this system: training, feature extraction, data gathering on flows, and network data categorization. This security system uses a number of machine learning models, such as decision trees, logistic regression, support vector machines, and nearest neighbors, to differentiate between legitimate and

malicious communications. The best algorithm is one that can identify attacks with the fewest false positives and the greatest accuracy. In their demonstration of an attack detection approach, *Chen et al. (2018)* used the XGBoost predictor. We took advantage of the controller's features to include the traffic gathering and categorization models. Finding the best way to deal with the controller's fears was the driving force behind the inquiry.

*Smys, Basar & Wang (2020)* introduced a hybrid approach that integrated traditional machine learning with a conventional technique to create an intrusion detection system for Internet of Things networks. This IDS model employs a two-step process to identify network intrusions. First, it trains CNN model utilizing features extracted by LSTM model. The CNN model subsequently use such characteristics to detect network breaches. The system shown in *Saba et al. (2022)* utilizes deep learning and anomaly detection techniques to provide an intrusion detection system specifically designed for the Internet of Things. An intrusion detection system must use a convolutional neural network to oversee all network traffic, including both inbound and outbound data connections. The primary objective of study (*Hassan et al., 2024*) was to promptly identify hazards within an SDN system. Despite being a next-generation network architecture, the SDN was susceptible to breaches due to its centralised configuration options.

## METHODOLOGY

The proposed intrusion detection system shown in Fig. 1 employs an innovative and automated approach to detect and categorize network intrusions using data from the IoT networks. We eliminate any missing values (NaN), unnecessary features, and redundant features to prepare the SDN intrusion data. We then partitioned the data into two sets, organized it, and systematically transformed it using a label encoder, standard scaler, and other relevant methods. We optimize deep models *via* parameter modification, utilizing both the proposed lightweight model for feature extraction and the extracted data for attack detection. We evaluate performance using multiple metrics.

### Dataset details

InSDN is an extensive dataset specifically designed for assessing intrusion detection systems in the context of SDN. The recently published dataset comprises a diverse array of assaults, including both harmless and harmful ones, that have the potential to target several standard components of SDN. InSDN considers a range of threats, including DoS, DDoS, brute force attacks, web application vulnerabilities, exploitation attempts, probes, and botnets. The created data is often sent over many widely used application services, such as email, File Transfer Protocol (FTP), Secure Shell (SSH), Domain Name System (DSH), Hypertext Transfer Protocol Secure (HTTPS), Hypertext Transfer Protocol (HTTP), Secure Sockets Layer (SSL), and others. We generated the dataset using four virtual machines. The first virtual computer serves as a substitute for the malevolent server running Kali Linux. The ONOS controller connects to the secondary system, an Ubuntu 16.4 server. The third server is an Ubuntu 16.4 system that is compatible with both OVS and Mininet switches. In addition, we used the same OVS technology to create the Damn

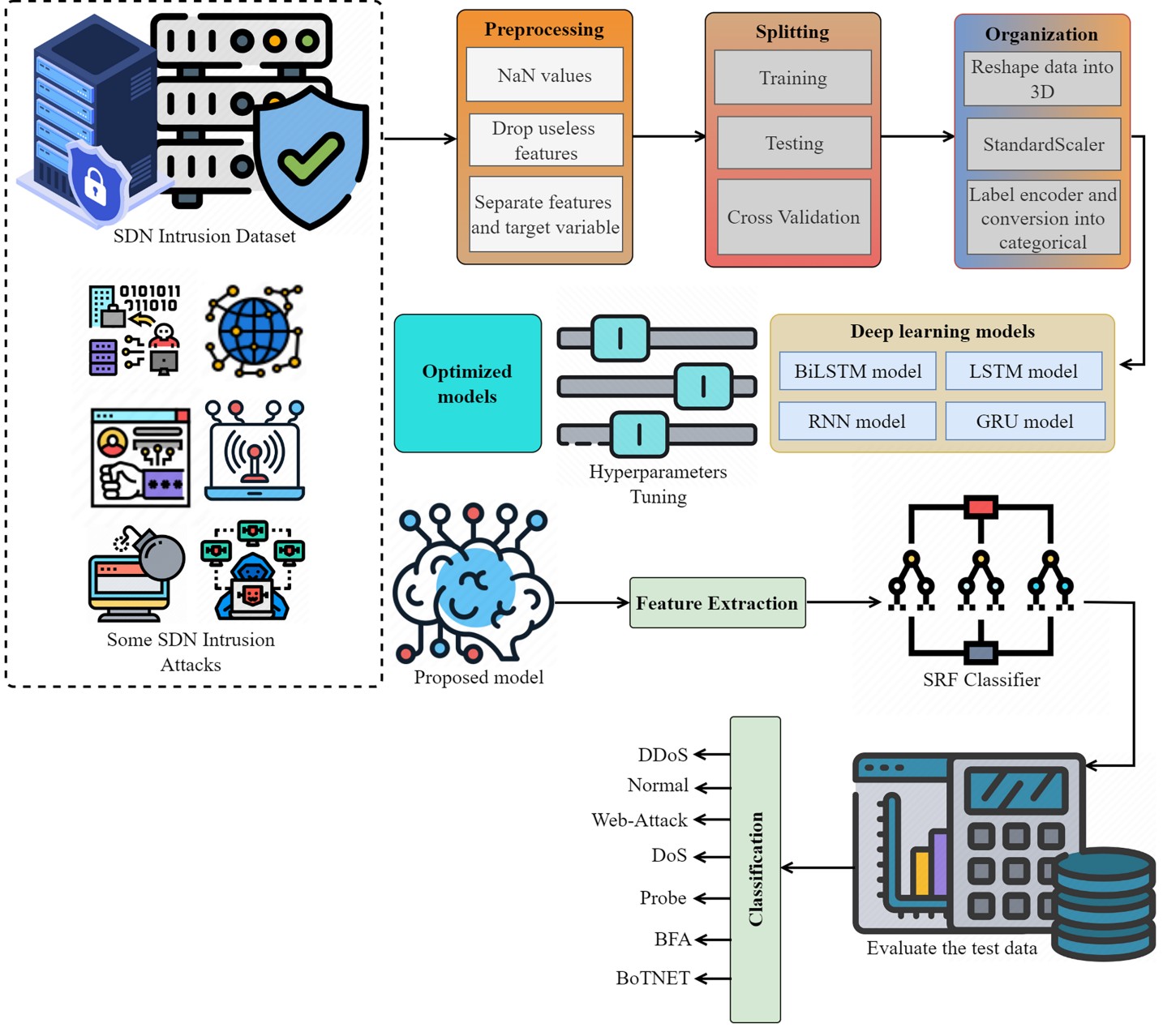

**Figure 1 The proposed methodology for SDN intrusion detection-based attacks is visualized, adhering to the detailed flow of the methodology.**

Vulnerable Web Application Server (DVWA) inside a Docker container. We also evaluated many potential attack scenarios that may arise from both inside and outside the SDN network. The OVS_Group is a fixed representation of the OVS server's assault recordings. The six components of the course address various topics like web assaults, brute force attacks, botnets, distributed denial of service, and probing attacks (*Badcodebuilder, 2023*). The dataset records are shown in Fig. 2.

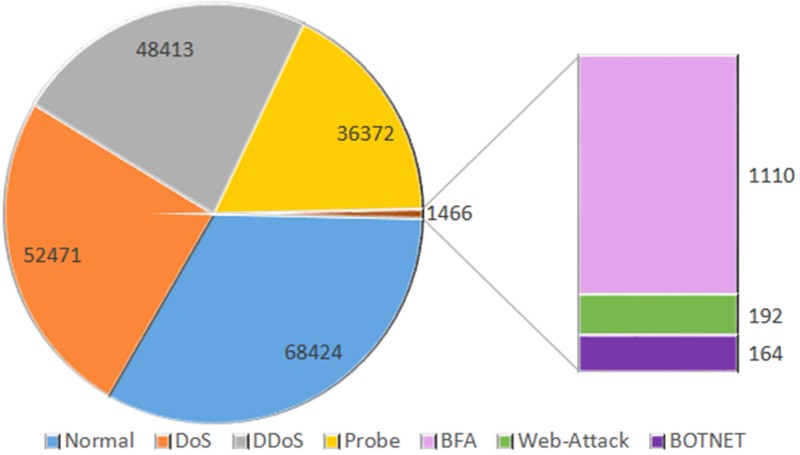

**Figure 2** Dataset size: the total number of counts of normal and OVS group data.

## Data preprocessing

Data preparation is the process of modifying the values in a dataset in order to improve data collection and manipulation. Because there is such a large discrepancy between the dataset's top and lowest values, normalizing the data makes the method go more smoothly. According to reference (*Yadav & Kalpana, 2019*), there are substantial advantages to data normalization for neural network algorithm classification. Training neural networks utilizing the back-propagation technique becomes much faster with input value normalization applied, resulting in a more efficient neural network. Dataset processing begins with removing null values and empty cells. To prevent any potential damage to the model, we eliminated rows with empty data. When training on massive datasets, it may not always be a good idea to use every feature. Some features of the dataset can influence the outcome negatively. By obtaining the dataset's correlation matrix, we may identify superfluous features and eliminate those with high correlation values (*Yang et al., 2022*).

## Reshape data

Data reshaping modifies the configuration of data from one format to another to enable analysis or subsequent processing. During this phase, rows and columns may be reorganized, the data type may be modified, or the format may shift from wide to long. The numpy.reshape() function can change the shape of a numpy array without modifying its data. To maintain compatibility of arrays with supplementary operations, it is often necessary to alter their geometry. The np.reshape() function modifies the array's shape and thereafter returns the array in its new configuration. The only requirement is that both the original array and the updated array possess the same number of elements.

## Normalization

Data designed for machine or deep learning is frequently normalized before applications. Normalization is the procedure of ensuring uniformity in the dimensions of all columns within a dataset. Normalizing each sample is not requisite for machine learning. It is

```
# Reshape data
if len(X_train.shape) == 2:
    X_train = np.reshape(X_train, (X_train.shape[0], 1, X_train.shape[1]))
    X_test = np.reshape(X_test, (X_test.shape[0], 1, X_test.shape[1]))

# Normalize the data
scaler = StandardScaler()
X_train = scaler.fit_transform(X_train.reshape(-1, X_train.shape[-1])).reshape(X_train.shape)
X_test = scaler.transform(X_test.reshape(-1, X_test.shape[-1])).reshape(X_test.shape)

# Encode the labels
label_encoder = LabelEncoder()
y_train = label_encoder.fit_transform(y_train)
y_test = label_encoder.transform(y_test)

# Convert labels to categorical
num_classes = len(np.unique(y_train))
y_train = to_categorical(y_train, num_classes=num_classes)
y_test = to_categorical(y_test, num_classes=num_classes)
```

**Figure 3 Code snippet for label encoding, reshaping of data, and normalization.**

crucial when the feature ranges vary. The use of normalized data enhances both the accuracy and utility of a model. A prevalent method of discussing min-max scaling is to achieve uniformity. Modification of the features occurs within a specific range, typically between 0 and 1. This is advantageous in scenarios where upholding the original range is essential and remains inside the specified range, allowing for the original figures to be comprehended.

### Label encoder

A prominent technique in artificial intelligence is to transform category data into a numerical representation that systems can readily supervise. An effective method for doing this is by label encoding, where each category item is assigned an integer value based on its alphabetical order. This strategy is direct and entails transforming every value in a column into a numerical representation (*Jia & Zhang, 2021*). The description of how to performed label encoder, reshape of data and normalization are shown in Snipped Fig. 3.

### Data splitting

To split the data into a training set and a testing set, the study used the (train_test_split)() function as illustrated in Fig. 4. It is necessary to categorize data into two distinct groups: attributes (X) and labels (Y). The dataset has four variables: $X_t rain$, $X_t est$, $y_t rain$, and $y_t est()$. Using the $X_t rain$ and $y_t rain$ datasets, perform model training and fine-tuning. Evaluate the model's output and label predictions by comparing them with the $X_t est$ and $y_t est$ sets. We have the option to conduct direct tests on the percentages of the training or test set. It is often advisable to ensure that training sets are larger than our test sets. A specific collection of data known as the training dataset shapes the model. The model receives instructions from the dataset. Based on its observations, the model learns new

```
# Separate the features (X) and the target variable (y)
X = data.drop('Label', axis=1)
y = data['Label']

# Split the data into training and testing sets
X_train, X_test, y_train, y_test = train_test_split(X, y, test_size=0.20, random_state=42)
```

**Figure 4 Data splitting, separate features and target label from the SDN intrusion dataset.**

knowledge (*Palimote, Atu & Osuigbo, 2021*). The import statements for the Pandas, Scikit-Learn, and Numpy libraries are imported. X encompasses the characteristics, whereas Y encompasses the labels. After dividing the dataframe into two equal halves, we performed a train-test split on both the X and Y halves. In order to ensure the consistency of data, the $random_state$ parameter functions as a seed for numpy.

## Proposed hybrid LSTM-SRF architecture

An LSTM utilizes ways to handle both short-term memory and long-term memory. LSTMs use gates to enhance and expedite the computational process. Recurrent neural networks (RNNs) trained using backpropagation are notorious for their struggles in learning long-term dependencies due to the issue of vanishing and exploding gradients. To mitigate the visibility of expanding gradients, one may use gradient cropping. However, addressing the issue of fading gradients requires a more intricate approach. The LSTM model proposed by *Hochreiter & Schmidhuber (1997)* was an early and successful solution to the problem of vanishing gradients. While LSTMs use memory cells instead of ordinary recurrent nodes, their behavior is largely similar to that of conventional recurrent neural networks. Each node in the network corresponds to the state of an internal memory cell. There is a single recurrent edge with a constant weight between this node and itself. Because of this design, the gradient may require many time steps to propagate in order to avoid becoming too small or large. Figure 5 presents the proposed deep extractor framework based on deep and supervised RF (random forest) technique for best performance in attack detection.

This concept gives rise to a "long, short-term memory". Weights can help simple recurrent neural networks preserve their memories over time. Training gradually adjusts the weights to collect and display a diverse range of data-related information. Transient activation, which transfers across nodes, contributes to their short-term memory. A memory cell serves as an additional storage component inside the LSTM architecture. The intricate configuration of a memory cell is determined by the arrangement of basic nodes interconnected in a certain way with multiplicative nodes. LSTMs specifically aim to reduce long-term dependence. The capacity to retain information over an extended period is mostly an inherent inclination rather than a skill that needs deliberate cultivation. The sequence of modules that make up a recurrent neural network share the same characteristics as any other neural network. The primary element of traditional RNNs is a basic repeating module consisting of a single tanh layer.

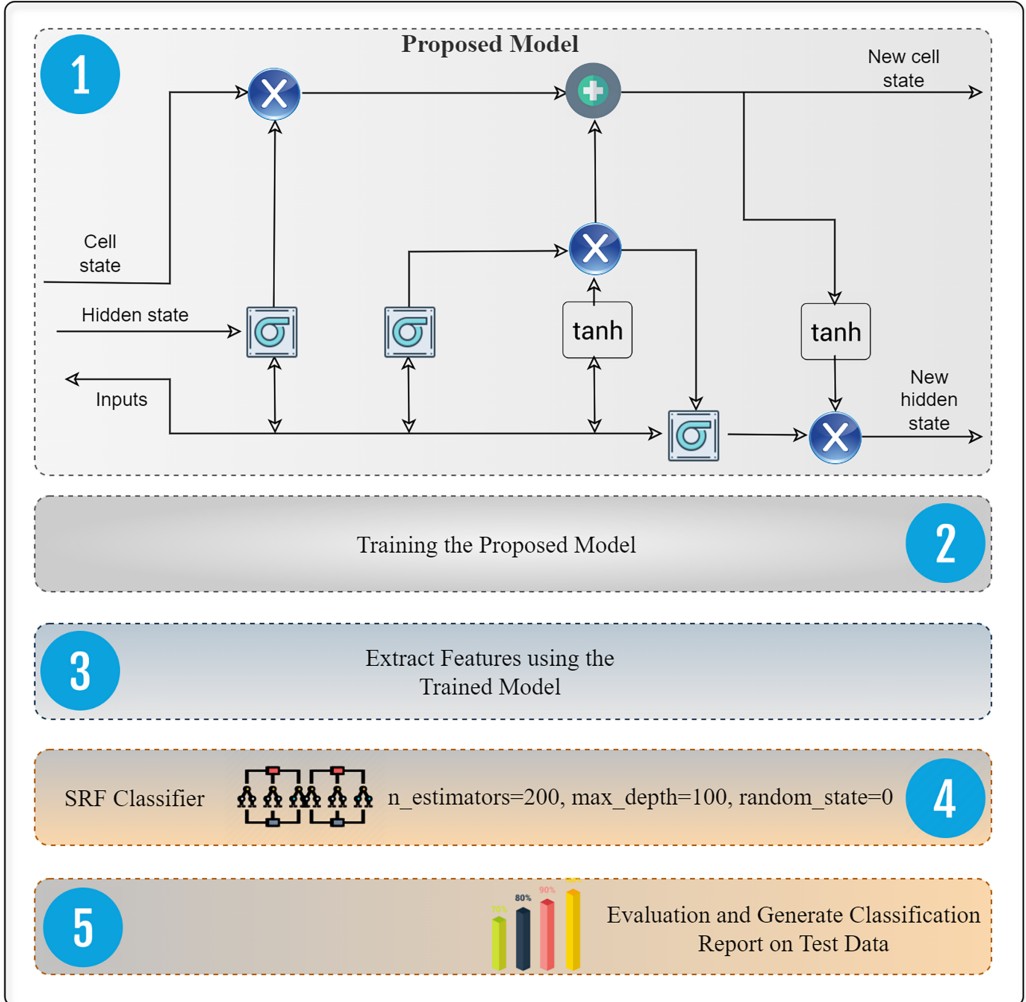

**Figure 5 Design of software defined network intrusion system that incorporates a deep feature extraction framework.** This system is designed to identify and detect malicious attacks specifically in IoT environments.               

### Input layer

The input layer is the first stage of the LSTM network. This results in continuous or sequential data where each data point is a vector representing the characteristics of the time series. By organizing the data in this way, the order and timing of sample events can be determined, which is important for applications such as anomaly detection.

### Hidden layer

It's also called LSTM layers, which are responsible for processing and storing data over time. Unlike traditional neural networks, these cells have an internal memory that uses structures called tags to determine what data to store, update, or retrieve at each stage. This feature allows the model to predict crucial sequences and effectively manage short- and long-term dependencies.

### Memory cell

Each LSTM acts as a mini-network unit. This type of cell maintains an internal state that changes between time domains, which helps the network "remember" important information and "forget" unimportant bits. It contains input, forget and output gates.

By passing *via* the forget gate, it eliminates superfluous data from the cell state. The gate processes the inputs $x_t$ (the input at the given time, x refers to the input and t for time) and $h_{t-1}$ (the output of the previous cell, $h$ refers to the hidden state and $t$ for time) by multiplying their weight matrices and adding the bias. An activation function processes the result into binary format. In certain cellular conditions, we exclude the data if the result is 0, but we retain it if the output is 1.

The input gate alters the cell's state by incorporating relevant data. To control the data, we use the sigmoid function to choose the values to save. Using $h_{t-1}$ and $x_t$ as inputs, it operates similarly to the neglect gate. Next, we use the tanh function to create a vector of all possible values for $h_{t-1}$ and $x_t$, which may take on values between −1 and +1. In order to acquire meaningful results, we multiply the vector values by the regulated values.

The output gate's primary function is to assess the cell's current condition and provide an output that contains valuable information. The first step involves using the cell's hyperbolic tangent function to generate a vector. Subsequently, we extract the desired values from the variables $h_t - 1$ and $x_t$ and subject them to the sigmoid function in order to control and refine the data. Before sending the vector and controlled values to the next cell, the last step is to perform a multiplication operation on them.

### Output layer

The output layer takes the subsequent information from the LSTM and converts it into a useful output. It typically consists of a single (fully connected) layer that feeds one or more hidden states to the target. The threat patterns such as DoS, DDoS, brute force attack, web applications, exploitation, probe, and botnet are analyzed. Furthermore, the normal traffic in the generated data covers various popular application services such as HTTPS, HTTP, SSL, DNS, Email, FTP, SSH, *etc.*

The supervised learning process uses the random forest method. The algorithm utilizes the bagging technique to train a group of decision trees, which together form the generated "forest." The bagging approach relies on the fundamental principle that using several learning models improves the final outcome. The main benefit of random forest is its capability to handle both classification and regression problems, which are the predominant types of tasks in contemporary machine learning systems. The random forest model becomes progressively more unpredictable as the number of trees increases. When splitting a node, it looks for the most beneficial trait from a randomly chosen collection of qualities, rather than the most important feature. This leads to a model that is typically better and exhibits a substantial degree of variety. Random forest streamlines the assessment of a variable's significance or impact on the model. Several techniques exist for assessing the significance of a trait.

**Table 1 Details about hyperparameters, activation functions, and optimizers for deep learning models.**

| Parameter | Value |
|---|---|
| Kernel regularizer l1 | 0.001 |
| Activation | Relu |
| Optimizer | Adam |
| Epochs | 100 |
| Batch size | 64 |
| Loss | Categorical |

## Models training

We constructed the proposed model and tested its intrusion detection performance using an Anaconda Jupyter Notebook on an Intel Core-i7 CPU with 16 GB of RAM and the Windows 10 environment. When we set the shuffle parameter to True, the shuffling process discloses our information. To begin the process, we use the $X_t rain$ and $X_t est$ datasets to train and optimize our model. Subsequently, we subjected it to rigorous testing to evaluate its performance. We used a 64-unit batch size, a 0.001 kernel regularizer, dropouts, and 100 epochs. Details about hyperparameters, activation functions, and optimizers for deep learning models is illustrated in Table 1.

## Evaluation

We evaluated the performance using several metrics such as the confusion matrix, ROC-AUC (*Sainis, Srivastava & Singh, 2018*), accuracy, precision, recall, F1-score, and precision-recall curves. We conducted an analysis and comparison of the proposed system's performance with that of other systems. The classifier only relies on the frequency of correct predictions as the sole statistic it evaluates. One way to measure accuracy is to calculate the ratio of correct predictions to the total number of forecasts made. A precision is one that yields the anticipated ratio of correct positive outcomes to the total number of actual positive outcomes. A label's recall is defined as the ratio of the total number of genuine positives to the number of true positives. The F1-score applies a more severe penalty to outliers. When the costs of false negatives (FN) and false positives (FP) are equal, the F1-score may serve as a reliable indicator for assessment. The true negative rate is noticeably higher.

$$Accuracy = \frac{IDTP + IDTN}{IDTP + IDFN + IDFP + IDTN} \tag{1}$$

$$Precision-PR = \frac{IDTP}{IDTP + IDFP} \tag{2}$$

$$Recall-RE = \frac{IDTP}{IDTP + IDFN} \tag{3}$$

$$F1\text{-}Score = 2 \times \frac{PR \times RE}{PR + RE}. \tag{4}$$

**Table 2  Classification performance using RNN method.**

|  | Precision | Recall | F1-score | TPS | WPS | Support |
|---|---|---|---|---|---|---|
| BFA | 0.8776 | 0.8595 | 0.8685 | 208 | 34 | 242 |
| BoTNET | 1.0000 | 1.0000 | 1.0000 | 29 | 0 | 29 |
| DDoS | 0.9989 | 1.0000 | 0.9994 | 9,655 | 0 | 9,655 |
| DoS | 0.9815 | 0.9987 | 0.9900 | 10,384 | 14 | 10,398 |
| Normal | 0.9996 | 0.9988 | 0.9992 | 13,887 | 16 | 13,903 |
| Probe | 0.9999 | 0.9775 | 0.9886 | 7,007 | 161 | 7,168 |
| Web-attack | 0.9444 | 0.4857 | 0.6415 | 17 | 16 | 35 |
| Accuracy |  |  | 0.9941 | – | – | 41,430 |
| Macro average | 0.9717 | 0.9029 | 0.6267 | – | – | 41,430 |
| Weighted average | 0.9942 | 0.9942 | 0.9941 | – | – | 41,430 |

# RESULTS AND DISCUSSION

This section provides the experimental results using RNN, GRU, bidirectional long short-term memory (BiLSTM), proposed LSTM and Hybrid model.

## Classification performance using RNN

To get the macro average of a particular metric, it is necessary to calculate the metric for all labels and then calculate the average without considering the percentage of each label in the dataset. To compute the average for a given set of labels, one must determine the metric on all label in the collection and then aggregate the number and percentage of all the label to get the weighted average of that metric. The performance of the deep RNN approach on the SDN dataset is shown in Table 2. All performance metrics are assessed in the experiments to evaluate the effectiveness and accuracy of the models in detecting network attacks. The BFA label achieved very poor scores in all categories. The BoTNET assaults achieved a performance of 100% utilizing accuracy, recall, and F1-score. Additionally, it made 29 positive predictions out of 29, with no erroneous predictions. The RNN approach attained an accuracy of 99.4% and a macro average of 97.1%.

## Classification performance using GRU

Table 3 gives an overview of how well the deep GRU technique performed on the SDN dataset. During the experiments, each and every performance measure is evaluated in order to determine how successful and accurate the models are in identifying network threats. In each and every category, the Web-Attack label received very poorly received evaluations. When it comes to accuracy, recall, and F1-score, the BoTNET attacks were able to attain a good performance. Additionally, it produced 29 accurate forecasts out of a total of 29, and it did not make any incorrect predictions. Using the GRU method, we were able to get an accuracy of 99.8% and a macro average of 95.2%.

## Classification performance using BiLSTM

Table 4 describes the results of the deep BiLSTM method on the SDN dataset. For the purpose of determining the models' efficacy and precision in detecting network threats, the

| Table 3 Classification Performance using GRU method. | | | | | | |
|---|---|---|---|---|---|---|
| | Precision | Recall | F1-score | TPS | WPS | Support |
| BFA | 0.9102 | 0.9628 | 0.9357 | 233 | 9 | 242 |
| BoTNET | 1.0000 | 1.0000 | 1.0000 | 29 | 0 | 29 |
| DDoS | 0.9998 | 1.0000 | 0.9999 | 9,655 | 0 | 9,655 |
| DoS | 0.9983 | 0.9986 | 0.9984 | 10,383 | 15 | 10,398 |
| Normal | 0.9996 | 0.9997 | 0.9997 | 13,899 | 4 | 13,903 |
| Probe | 0.9999 | 0.9979 | 0.9989 | 7,153 | 15 | 7,168 |
| Web-attack | 0.7586 | 0.6286 | 0.6875 | 22 | 13 | 35 |
| Accuracy | | | 0.9986 | – | – | 41,430 |
| Macro average | 0.9523 | 0.9411 | 0.9457 | – | – | 41,430 |
| Weighted average | 0.9986 | 0.9986 | 0.9986 | – | – | 41,430 |

tests assess all performance metrics. Reviews for the BFA-Attack label were mostly negative across the board. The accuracy, recall, and F1-score were all positively affected by the BoTNET assaults. We achieved a macro average accuracy of 94.1% and a precision of 99.5% by using the BiLSTM technique.

## Classification performance using proposed LSTM

Table 5 presents the results of the deep LSTM method applied to the SDN dataset. The DDoS attack obtained a precision rate of 99.98%, the BFA achieved a F1-score of 93.91%, the DoS achieved a F1-score of 99.93%, and the Web-assault earned a recall score of 54.29%. The LSTM model attained a macro recall of 93.26% and a F1-score of 94.34%. Additionally, method achieved weighted average performance of 99.90%.

## Classification performance using the proposed model

After the deep experiments, we see that LSTM method perfroms best, so we extract features from the leightweight LSTM network, and design supervised random forest model for the intrusion detection. Random Forest generates several bootstrap samples from the core dataset. Randomly swapping data points creates samples. The multiple subsets created by this approach may cause data differences in different trees. Decision trees do predictions by building on each other. Each tree puts in a "vote" for a class, and the class with the most votes obtains. Since each decision tree learns from a different data source, random forests may combine their predictions. Combination in prediction and voting in classification decrease tree variance in Random Forest, improving prediction accuracy. The results of the proposed method applied to the SDN dataset are shown in Table 6. The DDoS attack had an accuracy rate of 100%, the BFA had a recall score of 97.93%, the DoS had a precision score of 99.98%, and the Web-assault had a recall score of 82.86%. The model achieved a macro recall of 97.23% and a F1-score of 96.55%. In addition, the approach attained a weighted average performance of 99.93%.

There are four different multiclass classification methods: RNN, GRU, BiLSTM, and LSTM. Figure 6 depicts the training accuracy and loss graphs for each of these techniques. The results shown in Fig. 6 show that the LSTM outperformed the RNN in terms of the

**Table 4  Classification performance using BiLSTM method.**

|  | Precision | Recall | F1-score | TPS | WPS | Support |
|---|---|---|---|---|---|---|
| BFA | 0.5916 | 0.9876 | 0.7399 | 208 | 34 | 242 |
| BoTNET | 1.0000 | 1.0000 | 1.0000 | 29 | 0 | 29 |
| DDoS | 1.0000 | 0.9998 | 0.9999 | 9,655 | 0 | 9,655 |
| DoS | 0.9991 | 0.9855 | 0.9923 | 10,384 | 14 | 10,398 |
| Normal | 0.9995 | 0.9996 | 0.9996 | 13,887 | 16 | 13,903 |
| Probe | 0.9997 | 0.9987 | 0.9992 | 7,007 | 161 | 7,168 |
| Web-attack | 1.0000 | 0.6286 | 0.7719 | 17 | 18 | 35 |
| Accuracy |  |  | 0.9956 | – | – | 41,430 |
| Macro average | 0.9414 | 0.9428 | 0.9290 | – | – | 41,430 |
| Weighted average | 0.9972 | 0.9956 | 0.9960 | – | – | 41,430 |

**Table 5  Classification performance using proposed modified LSTM method.**

|  | Precision | Recall | F1-score | TPS | WPS | Support |
|---|---|---|---|---|---|---|
| BFA | 0.8951 | 0.9876 | 0.9391 | 239 | 3 | 242 |
| BoTNET | 1.0000 | 1.0000 | 1.0000 | 29 | 0 | 29 |
| DDoS | 0.9998 | 1.0000 | 0.9999 | 9,655 | 0 | 9,655 |
| DoS | 0.9994 | 0.9991 | 0.9993 | 10,389 | 9 | 10,398 |
| Normal | 0.9999 | 0.9996 | 0.9997 | 13,898 | 5 | 13,903 |
| Probe | 0.9999 | 0.9987 | 0.9993 | 7,159 | 9 | 7,168 |
| Web-attack | 0.8636 | 0.5429 | 0.6667 | 19 | 16 | 35 |
| Accuracy |  |  | 0.9990 | – | – | 41,430 |
| Macro average | 0.9654 | 0.9326 | 0.9434 | – | – | 41,430 |
| Weighted average | 0.9990 | 0.9990 | 0.9990 | – | – | 41,430 |

**Table 6  Classification performance using lightweight features SRF.**

|  | Precision | Recall | F1-score | TPS | WPS | Support |
|---|---|---|---|---|---|---|
| BFA | 0.9518 | 0.9793 | 0.9654 | 237 | 5 | 242 |
| BoTNET | 1.0000 | 1.0000 | 1.0000 | 29 | 0 | 29 |
| DDoS | 1.0000 | 1.0000 | 1.0000 | 9,655 | 0 | 9,655 |
| DoS | 0.9998 | 0.9992 | 0.9995 | 10,390 | 8 | 10,398 |
| Normal | 0.9999 | 0.9997 | 0.9998 | 13,899 | 4 | 13,903 |
| Probe | 0.9996 | 0.9993 | 0.9994 | 7,163 | 5 | 7,168 |
| Web-attack | 0.7632 | 0.8286 | 0.7945 | 29 | 6 | 35 |
| Accuracy | – | – | 0.9993 | – | – | 41,430 |
| Macro average | 0.9592 | 0.9723 | 0.9655 | – | – | 41,430 |
| Weighted average | 0.9993 | 0.9993 | 0.9993 | – | – | 41,430 |

accuracy of its training. An increase in the number of epochs led to an improvement in the accuracy of the models. The figures also feature loss graphs, which are included in order to provide a better understanding of the efficacy of the model.
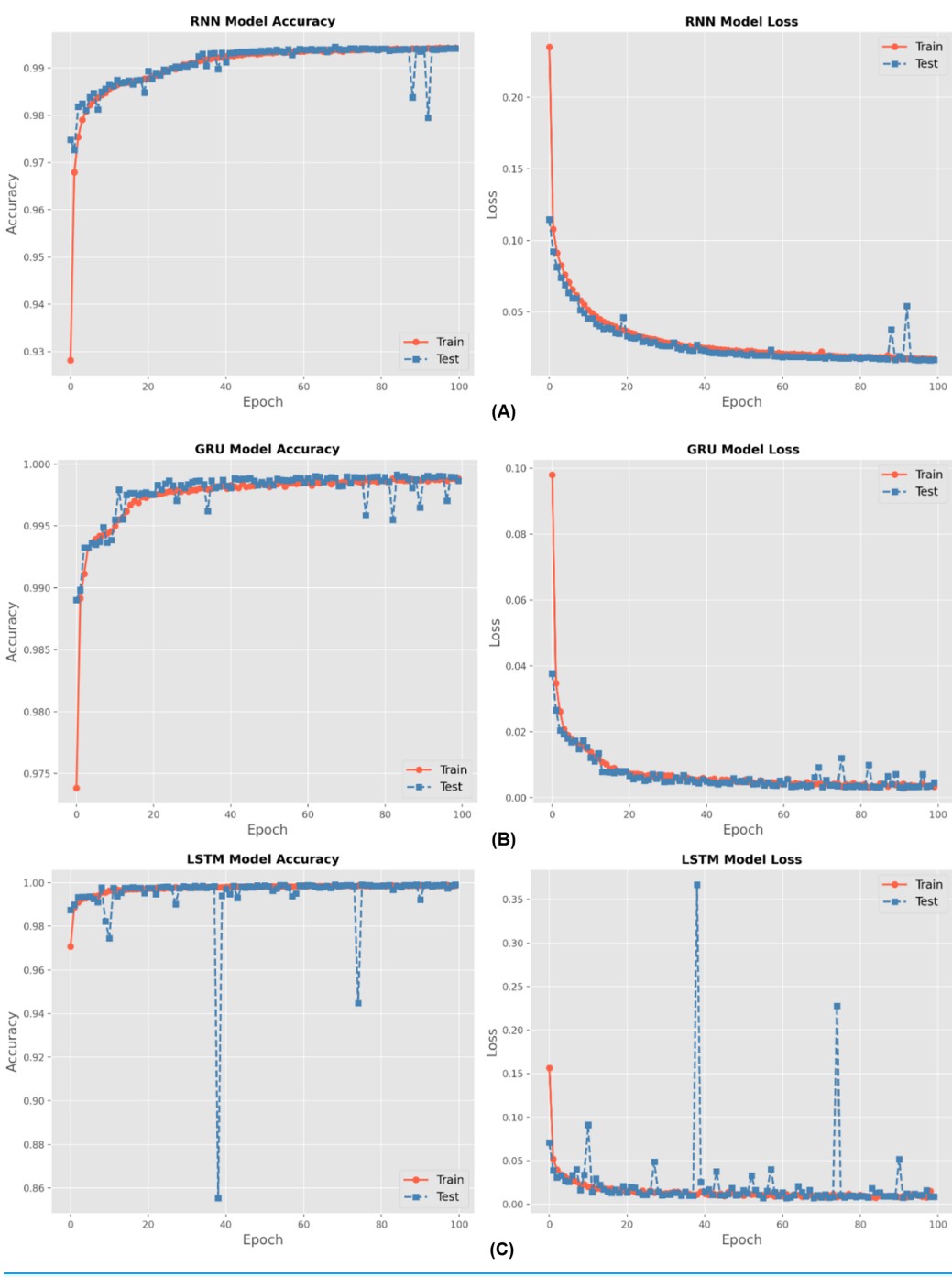

**Figure 6 Accuracy and loss for (A) RNN, (B) GRU and (C) LSTM methods during hundred epochs.**

Figure 7 displays the precision and recall curves for the multiclass classification of the proposed model architecture. The proposed approach successfully classified network connection records as either malicious or legitimate. The suggested strategy has shown to be successful for the majority of attack types in the multiclass category. This demonstrates

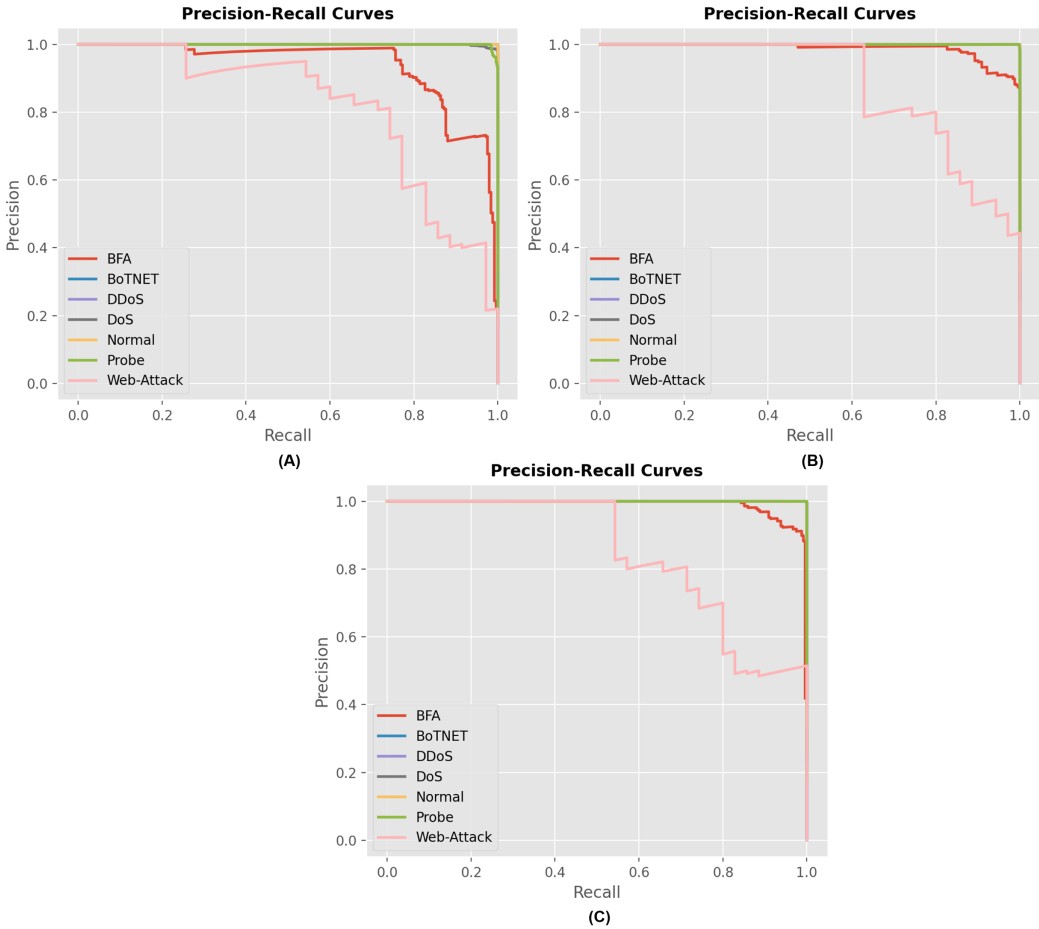

**Figure 7 Precision and recall curves for different model describing the performance of each attack for (A) RNN, (B) GRU and (C) LSTM model.**

the credibility of the proposed method and its ability to enhance the accuracy of multiclass classification.

## TSNE visualization

Deep learning models can consist of several hidden layers, which might be a challenge when it comes to understanding them. These models may be likened to a "black box," concealing the internal mechanisms by which they analyze incoming data in order to provide the most effective outcomes. In order to get a more comprehensive understanding of how deep learning models acquire knowledge from various hidden layers, a technique called t-distributed stochastic neighbor embedding (t-SNE) may be used for feature visualization (*Hamid & Sugumaran, 2020*). The t-SNE visualization method, similar to principal component analysis (PCA), reduces high-dimensional data to lower-dimensional values and displays them in two dimensions (*Chapagain et al., 2022*). We used features extracted from the penultimate layer of the proposed model for both binary and multiclass classification. Figures 8 and 9 depict the graphs for multiclass classification. These graphs illustrate the two dimensions of the inputs used in t-SNE. The clusters in which the assault

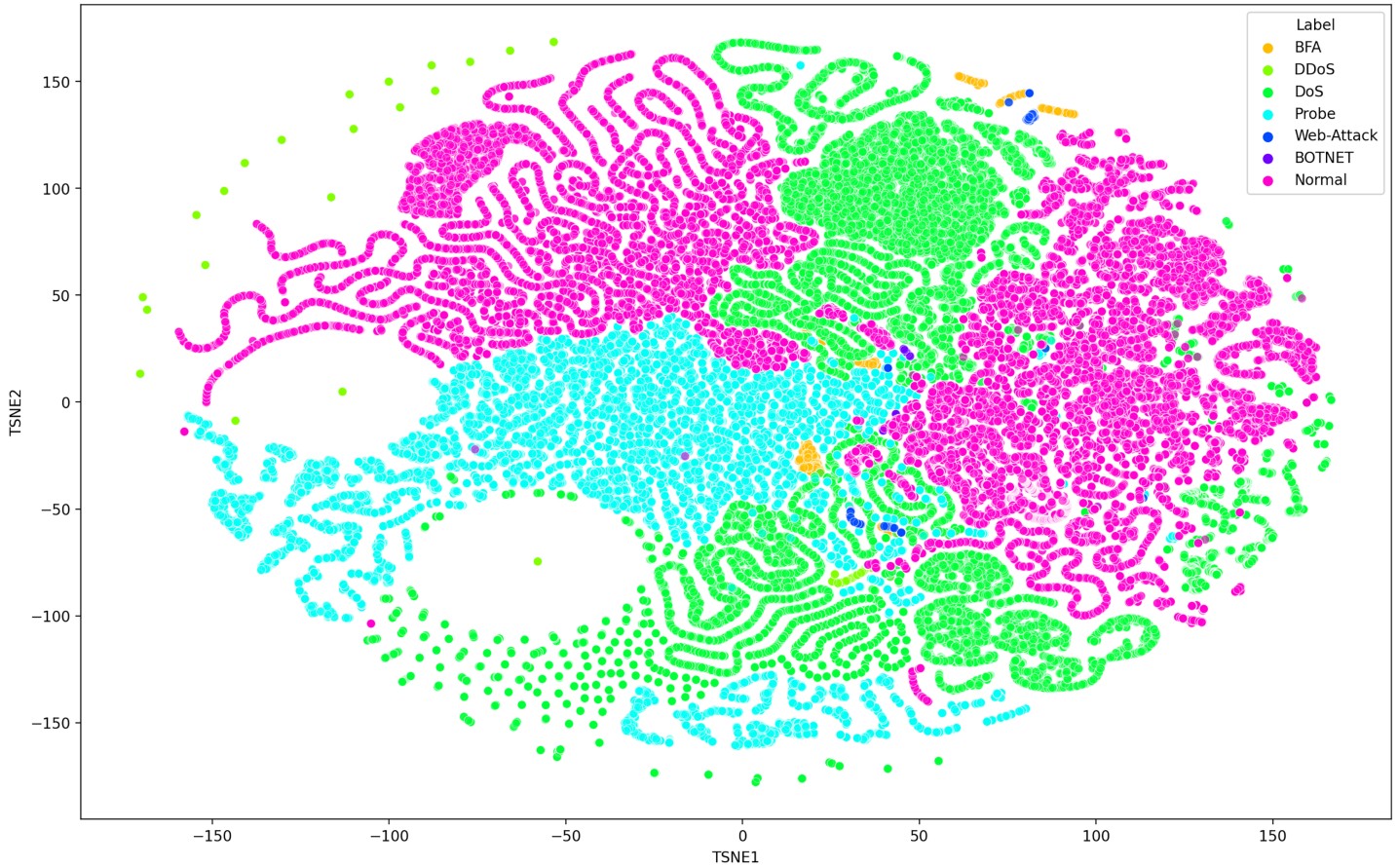

**Figure 8 The t-SNE visualization method, similar to principal component analysis (PCA), reduces high-dimensional data to lower-dimensional values and displays them in two dimensions using SDN dataset consists seven attacks.**

and typical traffic data occurred exhibited little overlap, as seen in Fig. 8. The model's ability to differentiate between malicious and benign traffic is clearly shown here. Figure 10 illustrates the classification of various assaults based on their distinct characteristics. Conversely, the model may have learned several attack-specific feature patterns, since most forms of assault divide the clusters into two or more.

## Confusion matrix

Confusion matrix are numerical arrays that highlight model performance. Classification models predict performance in each category (*Sokkalingam & Ramakrishnan, 2022*). Confusion matrix link results to initial data categories. This data shows that confusion matrix are beneficial in supervised learning when the output distribution is known. The confusion matrix simplifies classifier accuracy estimates for broad and specialized categories. It also helps identify additional essential factors that model developers use to evaluate the work. A confusion matrix may enable learning, which combines numerous models on the test set of the same dataset to provide the best results as seen in Fig. 10. Even with several classes, the confusion matrix ideas stay the same.

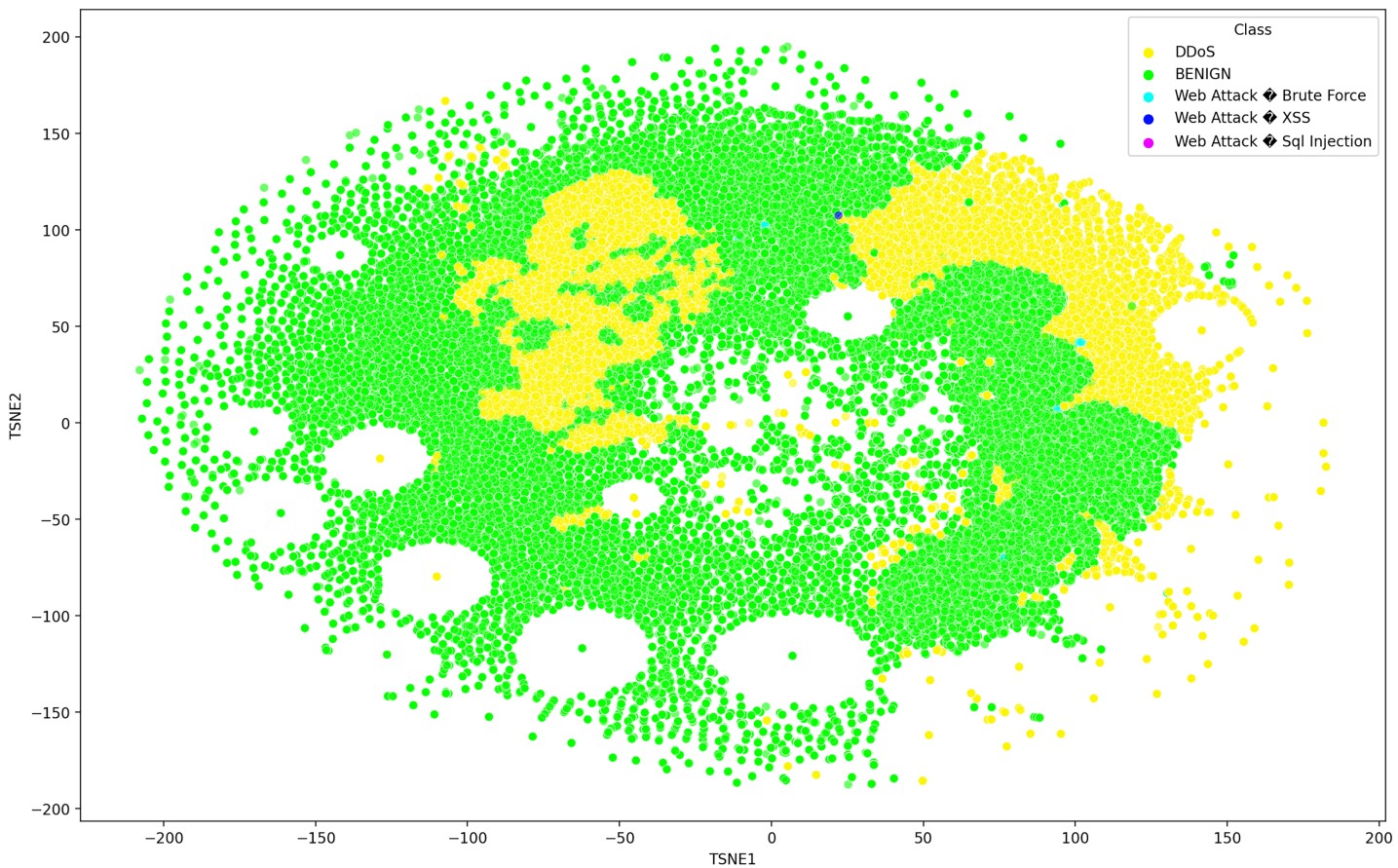

**Figure 9** The t-SNE visualization technique, reduces high-dimensional data to lower-dimensional values and exhibits them in two dimensions using the SDN dataset 2 the SDN dataset comprises five attacks.

## Experiments with additional dataset

We further evaluate the efficacy of the proposed model using additional multi-class datasets, each comprising five classes. Table 7 demonstrate that web-based attacks such as XSS and SQL injection earned a stunning precision score of 100%, while DDoS attacks recorded 99.98% and benign instances achieved 99.87%. Conversely, brute force attacks yielded a precision of only 46.09%. The DDoS attack achieved a 99.94% F1-score and a 95% weighted F1-score. The model demonstrated superior performance on an alternative multi-class dataset.

## Proposed model results using SMOTE technique

The results obtained from the balanced InSDN dataset using SMOTE technique is illustrated in Table 8. SMOTE is an abbreviation for the synthetic minority over-sampling technique. This methodology is an oversampling strategy that produces synthetic minority class samples to balance the dataset. SMOTE is a technique for augmenting data by employing interpolation between instances of minority classes to create new synthetic samples.

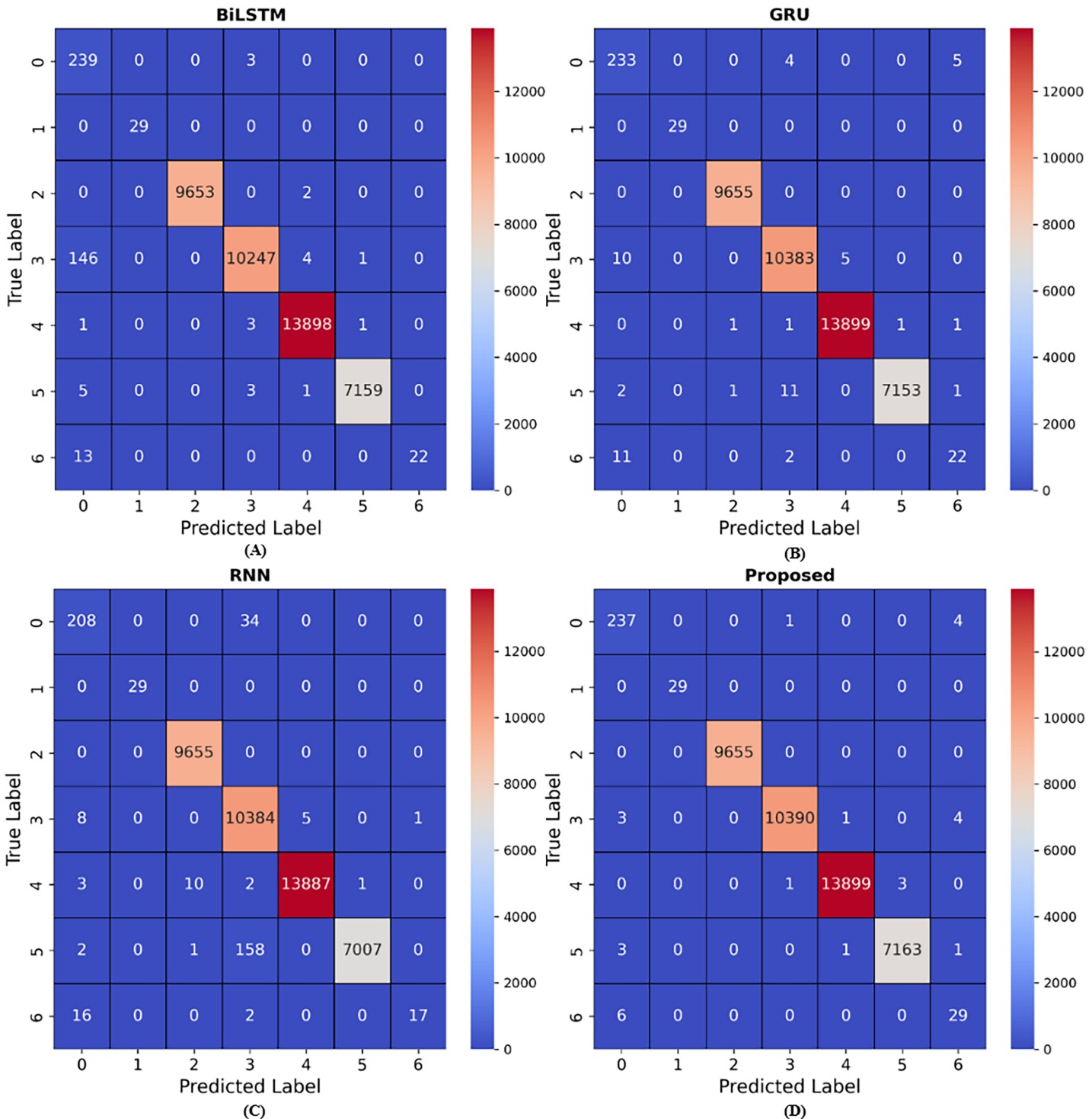

**Figure 10** **Results for confusion matrix.** (A) presents the confusion matrix results for BiLSTM model, (B) presents the confusion matrix results for GRU model, (C) presents the confusion matrix results for RNN model, (D) presents the confusion matrix results for Proposed model. The BiLSTM model achieved nine wrong predictions for class 5, GRU model achieved 15 wrong predictions, RNN model achieved 161 wrong predictions and the proposed model achieved five wrong predictions.

**Table 7 Proposed model results on additional multi-class datasets.**

| Attack | Precision | Recall | F1-score |
|---|---|---|---|
| Benign | 0.9987 | 0.9307 | 0.9635 |
| DDoS | 0.9998 | 0.9989 | 0.9994 |
| Web attack (brute force) | 0.4609 | 1.0000 | 0.6310 |
| Web attack (XSS) | 1.0000 | 0.5833 | 0.7368 |
| Web attack (SQL injection) | 1.0000 | 0.0317 | 0.0614 |
| Weighted avg. | 0.9763 | 0.9495 | 0.9500 |

**Table 8 Proposed model results using SMOTE technique.**

| | Precision | Recall | F1-score | TPS | WPS |
|---|---|---|---|---|---|
| BFA | 0.9878 | 1.0000 | 0.9938 | 242 | 0 |
| BOTNET | 1.0000 | 1.0000 | 1.0000 | 29 | 0 |
| DDoS | 1.0000 | 1.0000 | 1.0000 | 9,655 | 0 |
| DoS | 0.9995 | 0.9990 | 0.9993 | 10,388 | 10 |
| Normal | 0.9998 | 0.9994 | 0.9996 | 13,895 | 8 |
| Probe | 0.9993 | 0.9994 | 0.9994 | 7,164 | 4 |
| Web-attack | 0.8537 | 1.0000 | 0.9211 | 35 | 0 |
| Accuracy | | | 0.9995 | – | – |
| Macro average | 0.9771 | 0.9997 | 0.9876 | – | – |
| Weighted average | 0.9995 | 0.9995 | 0.9995 | – | – |

Synthetic samples are produced by SMOTE to create links among proximate neighbors in the feature space. SMOTE produces new minority class samples by repeatedly transitioning from a minority class sample to one of its k nearest neighbors in the feature space, with k being a parameter of the algorithm.

## Comparison with existing state of the art work

Table 9 presents a comparison with our research results and those of other scholars who have examined intrusion detection systems in SDN networks. ML/DL models and risk classification in SDN networks need the use of numerous datasets. *Ravi, Chaganti & Alazab (2022)* utilised the deep learning GRU approach for attack classification with the SDN dataset, attaining an accuracy of 98%. *Said, Sabir & Askerzade (2023)* employed a hybrid approach combining CNN and BiLSTM in an SDN environment, achieving an accuracy of 97%. *Arthi, Krishnaveni & Zeadally (2024)* also employed a hybrid model, specifically DNN-SVM, in an SDN context and attained 96% accuracy. *Tang et al. (2020)* utilised the NSL-KDD dataset for intrusion detection system (IDS) attack detection and attained a notably poor accuracy. *Chaganti et al. (2023)* used simple LSTM model for the classification of SDN attacks. The authors employed two datasets, one with two classes and the second with five classes, for the classification. The LSTM attained low accuracy and did not explain the preprocessing techniques well. Our proposed model developed a hybrid approach, LSTM-SRF, to extract features using deep learning and make detections using

**Table 9  Comparison with existing state of the art work.**

| Ref. | Method | Datasets | System | Accuracy |
|---|---|---|---|---|
| *Ravi, Chaganti & Alazab (2022)* | GRU | SDN | Intrusion detection | 98.4 |
| *Said, Sabir & Askerzade (2023)* | CNN-BiLSTM | SDN | Intrusion detection | 97.1 |
| *Arthi, Krishnaveni & Zeadally (2024)* | DNN-SVM | SDN | Intrusion detection | 96.7 |
| *Logeswari, Bose & Anitha (2023)* | HFS-LGBM | NSL-KDD | Intrusion detection | 98.7 |
| *Hadi & Mohammed (2022)* | CNN | NSL-KDD | Intrusion detection | 98.6 |
| *Tang et al. (2020)* | GRU-RNN | NSL-KDD | Intrusion detection | 89.1 |
| *Chaganti et al. (2023)* | LSTM | SDN | Intrusion detection | 97.1 |
| *Elsayed et al. (2021)* | CNN | SDN | Intrusion detection | 98.6 |
| (Our) | SRF | SDN | Intrusion detection | 99.9 |

the SRF model. Also, we utilized two datasets, one with five classes and the second with seven classes, which is different from the mentioned work. Our proposed model attained superior performance and uses fewer resources, making it computationally efficient. Unlike previous studies that used SDN networks for training and testing purposes, our research achieved a remarkable accuracy rate of 99% in classifying attacks.

## Explainable artificial intelligence

We employed explainable artificial intelligence (XAI) techniques such as SHAP. SHAP aims to enhance the comprehensibility and interpretability of AI models for humans. The objective of SHAP is to elucidate the mechanisms by which models make decisions, enabling professionals to comprehend the rationale underlying forecasts, classifications, and recommendations. These strategies facilitate our understanding of the prediction mechanisms employed by black-box models. Explainability of the proposed model using SHAP is shown in Fig. 11.

## Limitations and future work

Although the main focus of our article is on detecting IoT network intrusion attacks using deep learning, we ensure that it is crucial to integrate security methods with the SDN control layer to ensure robust security. To address such an issue in future work, we plan to introduce distributed SDN architectures to reduce the number of single-point nodes. Furthermore, the development of blockchain technology or security verification networks will improve the security of the regulatory platform.

The adversarial attacks seriously bypass deep learning-based models. Although our main objective was not adversarial, we will address this limitation in the future. Future research should also investigate adversarial examples and robust learning techniques to increase model resilience to new assumptions meant to evade detection.

We believe that one of the challenges facing the cyberthreats is the ever-changing nature of attacks. Such developments could make traditional IDS models obsolete over time. Although our study uses publicly available InSDN and other data sources, we ensure that continuous improvements are essential for long-term success. Future research will use customised training materials and federated learning techniques to adjust the model to

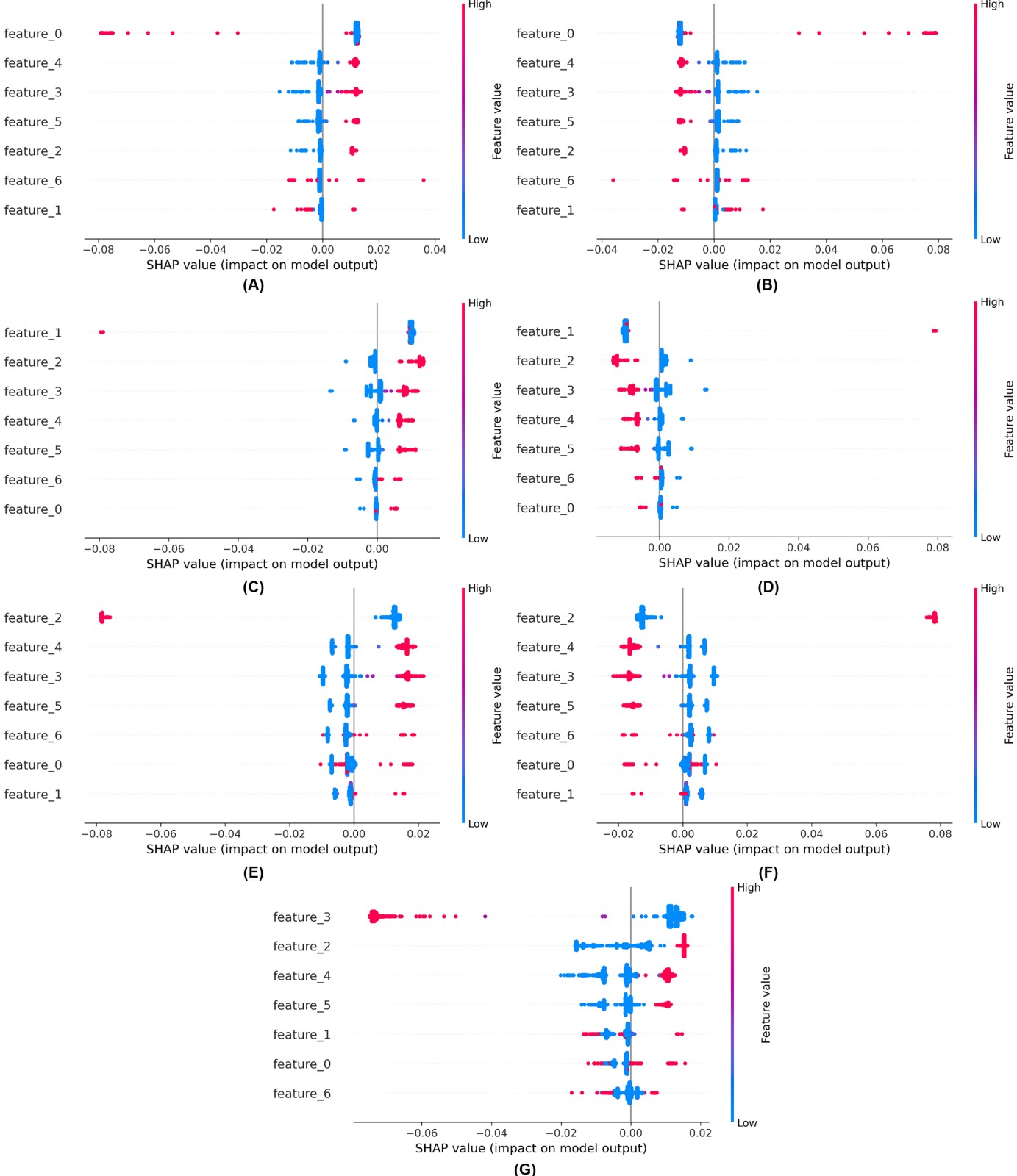

**Figure 11** Explainability of the proposed model using SHAP (A) for class 0, (B) for class 1, (C) for class 2, (D) for class 3, (E) for class 4, (F) for class 5, (G) for class 6.

new and developing threats. Additionally, we see potential in federated learning that allows multiple tools to collaborate on standardisation while also ensuring that systems remain up to date and privacy is maintained in the context of unauthorised data storage.

## CONCLUSION AND FUTURE DIRECTIONS

This article proposes a straightforward method to reliably detect attacks in intrusion-defined software-defined networking. We utilized an eleven-layer LSTM model to extract attacks from the dataset, subsequently feeding the extracted features into supervised random forests for various attack detections. Our comparative analysis demonstrated that the proposed model outperformed classifiers and showed superior capability in identifying different types of network attacks in the dataset, surpassing existing deep learning models such as RNN and GRU. By evaluating our model on an alternative dataset, we have also shown the applicability of our proposed approach to detecting intrusions in SDN networks as a whole.

The results indicate that the RNN model produced 16 erroneous predictions, the GRU model 13, the BiLSTM model 18, the LSTM model 16, and the proposed hybrid model 6, utilising the InSDN dataset for web attacks. In the context of DoS, the RNN model produces 14 erroneous predictions, the GRU 15, the BiLSTM 14, the LSTM nine, and the proposed hybrid model eight, utilising the InSDN dataset. Upon balancing the dataset using SMOTE, applied just to the training data, we recorded zero erroneous predictions for web attacks and four for probe attacks. In overall, with imbalanced data, we recorded 28 erroneous predictions across all attacks throughout testing, whereas with balanced data, we noted 22 erroneous predictions. The proposed approach performs effectively on both imbalanced and balanced datasets for attack detection.

### Funding
This research is supported by Princess Nourah bint Abdulrahman University Researchers Supporting Project number (PNURSP2025R346), Princess Nourah bint Abdulrahman University, Riyadh, Saudi Arabia. The AIDA LAB, Prince Sultan University, Riyadh Saudi Arabia supported the Article Processing Charges (APC) for this publication. The funders had a role in study design, data collection and analysis, decision to publish, or preparation of the manuscript.

### Grant Disclosures
The following grant information was disclosed by the authors:
Princess Nourah bint Abdulrahman University, Riyadh, Saudi Arabia: PNURSP2025R346.
The AIDA LAB, Prince Sultan University, Riyadh Saudi Arabia.

### Competing Interests
The authors declare that they have no competing interests.

## Author Contributions

- Muhammad Mujahid conceived and designed the experiments, performed the experiments, performed the computation work, prepared figures and/or tables, and approved the final draft.
- Abeer Rashad Mirdad conceived and designed the experiments, performed the experiments, performed the computation work, authored or reviewed drafts of the article, and approved the final draft.
- Faten S. Alamri conceived and designed the experiments, performed the experiments, prepared figures and/or tables, and approved the final draft.
- Anees Ara analyzed the data, prepared figures and/or tables, authored or reviewed drafts of the article, and approved the final draft.
- Amjad Khan analyzed the data, prepared figures and/or tables, authored or reviewed drafts of the article, and approved the final draft.

## Data Availability

The InSDN: A Novel SDN Intrusion Dataset is available at Kaggle: https://www.kaggle.com/datasets/badcodebuilder/insdn-dataset/data

DOI: 10.1109/ACCESS.2020.3022633

The SDN Intrusion Detection is available at Kaggle: https://www.kaggle.com/datasets/subhajournal/sdn-intrusion-detection.

The code is available in the Supplemental File.

## Supplemental Information

Supplemental information for this article can be found online at http://dx.doi.org/10.7717/peerj-cs.3103#supplemental-information.

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
