# Peer review of "Software defined network intrusion system to detect malicious attacks in computer Internet of Things security using deep extractor supervised random forest technique"

_PeerJ Computer Science, doi:10.7717/peerj-cs.3103_

## Round 0.1 · original submission · Major Revisions

This manuscript requires in-depth improvement and clarification in regarding:
i) a clear statement and justification of the work's novelty:
How does this work advance beyond existing LSTM-based detection models?
What unique approach is taken in hybridizing LSTM, RNN, and GRU?
How does it outperform other models in accuracy, efficiency, or adaptability?

ii) a better highlight of the main scientific contribution:
Instead of stating evaluation as a contribution, highlight specific improvements (e.g., reduced false positives, improved detection speed).
Justify why the new dataset improves threat detection—does it include previously unseen attacks? Is it larger or more diverse than existing datasets?
Addressing Challenges in Detection Models:

iii) give more citations for the existing gaps (e.g., congestion, computational inefficiency, adaptability issues),

And clearly define how the proposed approach mitigates these limitations.

·

Basic reporting

1. The centralized control plane in SDN is vulnerable to attacks, potentially compromising the entire network.
2. The deep learning models require significant processing power and memory, making real-time intrusion detection challenging.
3. Minority attack classes may be underrepresented, leading to biased models.
4. Choosing relevant features from high-dimensional datasets is difficult and impacts model accuracy.
5. The approach might not perform well on large-scale or highly dynamic networks.

Experimental design

1. The IDS may face delays in detecting attacks, reducing its effectiveness in real-time environments.
2. Attackers could craft adversarial examples to bypass detection models.
3. Deep learning models used in intrusion detection are often black-box systems, making it difficult to interpret decisions.
4. The system's performance might drop when applied to unseen network traffic or different datasets.
5. The system may incorrectly classify benign traffic as malicious, leading to unnecessary interventions.

Validity of the findings

1. The proposed method may not work well across all network configurations and attack patterns.
2. Lightweight IDS solutions are necessary, as IoT devices have limited processing power.
3. Cyber threats constantly evolve, requiring frequent model updates and retraining.
4. DPI-based IDS methods struggle with encrypted traffic, making some attacks undetectable.
5. Without standardized benchmark tests, comparing IDS models across studies is difficult.

Additional comments

1. The need for extensive data preprocessing (label encoding, normalization, etc.) adds to the system's overhead.
2. The deep learning model may perform well on known data but fail to generalize in real-world scenarios.
3. Analyzing every packet in a high-speed network can cause delays and affect network performance.
4. IDS systems handling personal or sensitive data must comply with privacy regulations.
5. Combining LSTM with supervised random forests increases implementation complexity and maintenance requirements.

Reviewer 2 ·

Basic reporting

The report is clear and professional English are used

Explained the contribution of the research in the abstract. The statement that a lightweight method is too general.

The author can improve the problem statement and justification:

Clarify the Challenges: The statement about challenges is too broad. The author should specify the exact issues with existing detection and classification models, supported by citations from relevant research.

Use Citations: If claims about challenges or gaps in current models are made, they should be backed up with references to existing literature or empirical studies.

Refine Line 85-86: The statement in these lines appears to be vague or overstated. The author should rephrase it to make it precise and well-supported.

Clearly Define the Problem Statement: The document should explicitly state what problem the proposed LSTM-based model is solving. Is it improving accuracy? Reducing false positives? Enhancing real-time threat detection?

1. Clarify "NIDS" (Line 90)
If "NIDS" refers to Network Intrusion Detection System, it should be written in full the first time it appears in the text.
A brief explanation of its relevance to the research should be provided.
2. Improve Objective 1
The phrase "excessive method congestion" is unclear and should be discussed earlier in the Challenges section.
The author should explain why existing models cause congestion and how the proposed method reduces it (e.g., through efficiency improvements).
3. Refine Objective 2 (New Dataset Usage)
The author should clarify:
What makes this dataset novel?
Why is it chosen over other datasets?
How does it improve detection compared to existing datasets?
4. Revise Objective 3 (Evaluation Is Not a Contribution)
Evaluation itself is a standard step in research, not a contribution.
Instead, the contribution could highlight new evaluation metrics, improved accuracy, or benchmarking against existing models.

Experimental design

Clarify IoT Network Data Source: The document should provide a detailed explanation of the IoT network, including:

What type of IoT devices are involved?
How data is collected from the network?
The nature of the traffic or threat patterns being analyzed.
Explain the SDN Dataset:

Specify the dataset name and its source (publicly available or private).
Describe the type of data (e.g., traffic logs, packet headers, flow features).
Provide the dataset size and details about preprocessing, if applicable.
Expand Acronyms (Line 278): "RF" should be fully spelled out (Random Forest) the first time it appears in the text.

Clarify Input Variables (Line 290):

The input variables should be italicized for emphasis.
Explain why these two variables are selected and how they impact the LSTM model.
Provide a structured breakdown of LSTM layers (input layer, hidden layers, memory cells, output layer).
Clearly differentiate this work from Chaganti et al. (2023)—highlighting how the approach here improves upon Literature Review(LR).
Explain RNN-GRU Hybridization:

Describe how RNN and GRU are combined in the model.
Justify why this hybrid approach is chosen over using only LSTM, RNN, or GRU individually.
Deep Feature Extraction:

Specify the deep learning techniques used for feature extraction.
Clarify if feature extraction is automated or involves manual selection.
Explain how deep features improve detection and classification accuracy.
Make the Methodology More Specific: The current methodology is too general. It should include:

A step-by-step explanation of data preprocessing, model training, and evaluation.
Clear details about hyperparameters, activation functions, and optimization methods used.

Validity of the findings

results are accessed and compared.

Additional comments

Clearly Define Research Contributions
The contributions should be concrete and unique.

---

## Round 0.2 · Minor Revisions

Analysing the revised version by myself, I found that critics:

ii) a better highlight of the main scientific contribution still lacks more information. In particular in the lines 108-121 I noticed:
- bullet #2 is not really a contribution
- bullet #3 has a statement ("The proposed approach performs effectively on both imbalanced and balanced datasets for attack detection") which is not backed by corresponding data/graphs shown in the paper and convincing discussions.

Therefore, I suggest that the authors carefully rewrite the 'major contributions' subsection, sticking precisely and concretely to what their experimental results have shown.

---

## Round 0.3 · accepted · Accept

After revising and complementing the section about the main contributions, I am confident that the work is original, is of relevance and is suited for publication in PeerJ Computer Science. Congratulations!